# Confined Cu-OH single sites in SSZ-13 zeolite for the direct oxidation of methane to methanol

Hailong Zhang[1,2], Peijie Han[1], Danfeng Wu[1], Congcong Du[1], Jiafei Zhao[1], Kelvin H. L. Zhang[1], Jingdong Lin[1], Shaolong Wan[1], Jianyu Huang[3], Shuai Wang[1]✉, Haifeng Xiong[1]✉ & Yong Wang[4]✉

The direct oxidation of methane to methanol (MTM) remains a significant challenge in heterogeneous catalysis due to the high dissociation energy of the C-H bond in methane and the high desorption energy of methanol. In this work, we demonstrate a breakthrough in selective MTM by achieving a high methanol space-time yield of 2678 mmol molCu−1 h−1 with 93% selectivity in a continuous methane-steam reaction at 400 °C. The superior performance is attributed to the confinement effect of 6-membered ring (6MR) voids in SSZ-13 zeolite, which host isolated Cu-OH single sites. Our results provide a deeper understanding of the role of Cu-zeolites in continuous methane-steam to methanol conversion and pave the way for further improvement.

The quest for a more effective and economical utilization of methane, a globally abundant energy resource, has gained worldwide attention. Direct oxidative conversion of methane to methanol, a promising reaction in heterogeneous catalysis, offers the potential to transform methane into highly valuable chemicals and condensable energy carriers under mild operating conditions[1–3]. However, this reaction remains a formidable challenge in heterogeneous catalysis due to the high stability of the C-H bond of methane, with a dissociation energy of 435 kJ/mol, and the thermodynamically favored over-oxidation of methane. This makes the direct, selective conversion of methane to methanol extremely difficult. To achieve efficient catalytic conversion, it is critical to address the challenge through the synthesis of a suitable catalyst that can reduce the activation energy for C-H bond breaking and provide high selectivity towards methanol.

Inspired by the natural methane monooxygenase (MMO), which acts as a catalyst for the selective conversion of methane to methanol under aerobic conditions[4,5], copper ion-exchanged zeolites have been widely reported as highly-efficient catalysts for catalyzing the methane-to-methanol (MTM) reaction via either stepwise or continuous processes[6–16]. Recently, researchers have reported the direct conversion of methane to methanol over Cu-based catalysts using water as a mild oxidant[3,6,7]. This approach not only suppresses the over-oxidation of methane but also promotes the desorption of methanol from the catalyst surface. The successful demonstration of continuous catalytic oxidation of methane to methanol using water as the oxygen source[8] has sparked significant interest in the field of continuous MTM conversion under anaerobic conditions[15].

The proposed catalytic centers for the conversion of methane to methanol include Cu-oxo clusters, such as $[Cu_2(\mu\text{-}O)]^{2+}$, $[Cu_2(\mu\text{-}O)_2]^{2+}$, $[Cu_3(\mu\text{-}O)_3]^{2+}$[1,17,18], and single Cu sites, such as $[Cu\text{-}OH]^+$[9,19]. These active sites are typically located within the 8-membered-ring (8MR) voids of aluminosilicate zeolites, such as MOR and CHA-type zeolites. Studies have shown that the catalytic activity of these sites is influenced by their coordination structures and confinement environments within zeolite topologies[20,21]. For example, the different Cu-O-Cu angles of $[Cu_2(\mu\text{-}O)]^{2+}$ sites caused by varying confinement environments result in different confinement effects in activating the C-H bond of methane[21].

However, the potential of smaller voids, such as 6MRs, as hosts for Cu-based active sites has been largely overlooked. In this work, we report that the single Cu sites confined within the 6MR void of CHA zeolite exhibit a high methanol space-time yield of 2678 mmol $mol_{Cu}^{-1}$

[1]State Key Laboratory of Physical Chemistry of Solid Surfaces, College of Chemistry and Chemical Engineering, Xiamen University, 422 South Siming Road, Xiamen 361005, China. [2]College of Carbon Neutrality Future Technology, Sichuan University, Chengdu 610064, China. [3]Clean Nano Energy Center, State Key Laboratory of Metastable Materials Science and Technology, Yanshan University, Qinhuangdao 066004, China. [4]Voiland School of Chemical Engineering and Bioengineering, Washington State University, Pullman, WA 99164, USA. ✉e-mail: shuaiwang@xmu.edu.cn; haifengxiong@xmu.edu.cn; wang42@wsu.edu

h⁻¹ with 93% selectivity in continuous methane-steam conversion at 400 °C. The strong spatial confinement effect of the 6MR void, as determined through theoretical calculations, is responsible for this high catalytic activity. This study provides a clearer understanding of the role of Cu single sites confined in 6MR voids[17,22–24] and contributes to the improvement of the conversion of methane to methanol processes.

## Results

### Formation and identification of a copper single site in SSZ-13 zeolite

To prepare a single type of copper single site confined in only 6MR in CHA zeolites via ion exchange method, three main points during Cu ion-exchange procedure have to be considered, namely, (i) the pH values of Cu ion-exchanged solution must be well controlled to avoid the precipitation of copper cations and the non-exchanged copper ions in zeolite micropores must be removed via centrifugation and rinsing steps, since the precipitate and non-exchanged copper ions are the precursor of CuO nanoparticles; (ii) the amount of Cu cations exchanged to the zeolite framework must be well controlled to ensure the copper ions only entering into and occupying 6MRs, which as smaller rings than 8MRs have stronger electrostatic interaction with copper cations, this makes the copper cations preferentially exchanged to 6MRs and then to 8MRs[25]; (iii) a thermodynamically stable coordination structure of the exchanged copper ions must be formed in the confined zeolite rings. CHA-type zeolite (Si/Al = 10) was selected as a host for the synthesis of 6MR-confined copper single sites, because it has a simple and stable framework structure only consisting of 6MRs and 8MRs. The copper ion exchange was performed in $Cu(NO_3)_2$ solution with pH of 5.0-5.8 and under stirring in water bath at

80 °C. By this method, the copper ion-exchanged Cu/SSZ-13 precursors with different Cu/Al ratios were synthesized via varying $Cu^{2+}$ concentrations and reaction time. Subsequently, the centrifugation and rinsing steps were carried out to remove the non-exchanged copper cations. The final samples were dried via a slow step-drying process under vacuum condition from 40 °C to 120 °C, such a drying process is key to avoid the impact of fast desorption of water molecules from hydrated copper ions on the well-exchanged sites. The active Cu-exchanged SSZ-13 catalysts were obtained by calcination in static air at 550 °C, with Cu/Al = 0.05-0.22 (Supplementary Table 1). Powder X-ray diffraction of these samples shows typical diffraction peaks assigned to the CHA-type zeolite, and no crystalline phases of CuO or $Cu_2O$ were detectable (Supplementary Fig. 1). A slight decrease in specific surface areas and micropore sizes with the increase of Cu/Al was found based on $N_2$ adsorption-desorption isotherms (Supplementary Table 1).

The coordination structure and confinement environment of Cu active sites located in SSZ-13 zeolite were first investigated. FTIR spectroscopy in the T-O-T (T: Si or Al) bond vibration region is commonly used to differentiate the two types of single copper species, which are confined on 6MRs and 8MRs of CHA zeolite, respectively. As shown in Fig. 1a, the two perturbed IR bands appearing at ~900 and ~950 cm⁻¹ are assigned to bare Cu(II) single atoms on 6MRs and monocopper(II) Cu-OH sites on 8MRs, respectively. The assignment of the two perturbed bands is reasonable based on theoretical simulations (Supplementary Fig. 2) and the previous works[26–29]. The different confinement-coordination of the exchanged Cu(II) cations in CHA zeolites usually play an important role in perturbing the T-O-T vibration. Most researchers agree that the bare copper(II) cation charge-balanced by two framework Al sites on 6MR has stronger interaction

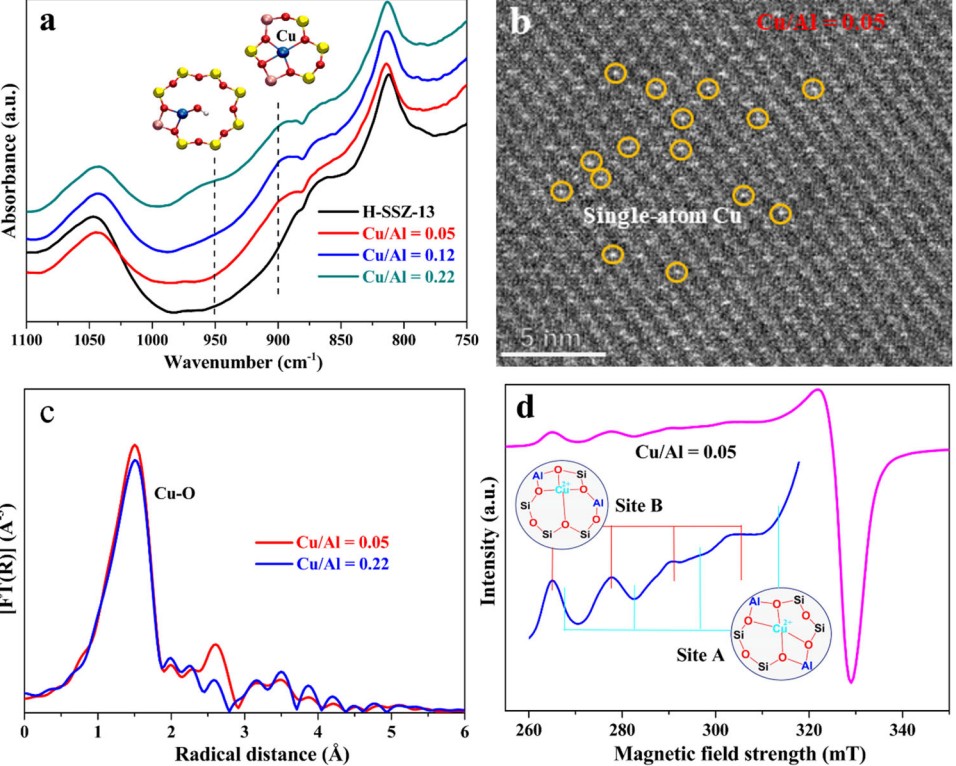

**Fig. 1 | Characterization of Cu₁/SSZ−13. a** FTIR spectra of Cu ion-exchanged SSZ-13 catalysts with Cu/Al ratios between 0.05 to 0.22. **b** HAADF-STEM image of Cu₁/SSZ−13 with Cu/Al = 0.05. **c** EXAFS results of the Cu/SSZ-13 samples with Cu/Al = 0.05 and 0.22. **d** EPR spectrum of Cu₁/SSZ-13 catalyst (Cu/Al = 0.05). The isolated bright spots (orange cycles) in (**b**) represent Cu single atoms, site A and site B in (**d**) indicate the two types of coordination structures of single Cu cation hosted in 6MRs.

with the zeolite framework than Cu-OH single site near one Al site on 8MR, thus the former shows a lower vibration wavenumber[26,29]. The bare Cu(II) single-atoms on 6MRs are found in all Cu/SSZ-13 catalysts (Cu/Al = 0.05-0.22). In particular, the one with Cu/Al ratio of 0.05 shows the bare Cu(II) single site on 6MR exclusively as evidenced by the sole appearance of the IR band at ~900 cm$^{-1}$, and is denoted hereinafter as Cu$_1$/SSZ-13. With the increase of Cu/Al ratios, the exchanged Cu cations started to occupy 8MRs and formed Cu-OH sites, which are identified by the variation of IR vibration at ~950 cm$^{-1}$ (Fig. 1a). HAADF-STEM images (Fig. 1b, Supplementary Fig. 3) directly demonstrate the sole presence of copper single-atoms on Cu$_1$/SSZ-13 based on the observation of isolated bright spots (orange cycles). Besides, the 6MR-hosted bare Cu single-atom sites on Cu$_1$/SSZ-13 are also evidenced by H$_2$ temperature-programmed reduction (H$_2$-TPR) and diffuse reflectance UV-Vis spectroscopy, which show a sole reduction peak at ~390 °C (Supplementary Fig. 4) and a featured absorption signal at ~13600 cm$^{-1}$ (Supplementary Fig. 5), both are assigned to bare Cu(II) single sites on 6MRs[17,26]. In contrast, Cu-oxo clusters (e.g., [Cu$_2$($\mu$-O)$_2$]$^{2+}$ and [Cu$_2$($\mu$-O)]$^{2+}$) and monocopper Cu-OH sites on 8MR of SSZ-13 were found at higher Cu/Al ratios (i.e., 0.12 and 0.22), as evidenced by the UV-Vis bands at 22200-35000 cm$^{-1}$ and H$_2$ reduction peak at ~250 °C, respectively[17,26]. Figure 1c shows Fourier transformed extended X-ray absorption fine structure (EXAFS) spectra of the Cu/SSZ-13 catalysts (Cu/Al = 0.05 and 0.22) in R space. The fitting results (Supplementary Fig. 6 and Table 2) reveal the average coordination number (CN) of the Cu species in these samples decreases from 4.02 to 3.49 as the Cu/Al ratio increases from 0.05 to 0.22, whereas their oxidation states remain at +2 (Supplementary Fig. 7). The combination of electron paramagnetic resonance (EPR) characterization and theoretical calculations further indicates that the tetra-coordinated bare Cu(II) atoms are mainly bound to three O atoms including two neighboring Al sites and one O atom between two adjacent Si sites of the 6MR (i.e., Site B in Fig. 1d. Charge density difference calculations are shown in Supplementary Fig. 8)[30–32]. The details on the characterization analysis can be found in the Supplementary Information.

## Catalytic activity for continuous methane-steam to methanol

Figure 2a shows the space-time yield (STY) of methanol in continuous methane-selective conversion to methanol over Cu$_1$/SSZ-13 catalyst or 6MR-confined Cu single sites at reaction temperatures ranging from 200 to 500 °C. The methanol STY exhibits a "volcano" trend with increasing reaction temperature with the highest methanol STY of 2678 mmol mol$_{Cu}^{-1}$ h$^{-1}$ (218 μmol g$_{cat.}^{-1}$ h$^{-1}$, Supplementary Fig. 9) and 93% selectivity achieved at the optimal reaction temperature of 400 °C. Such STY value is more than three times higher than that reported in recent literature[9] and even over an order of magnitude higher than most of previous reports on Cu-zeolites in a continuous process (Supplementary Table 3). Elevating reaction temperature to 450–500 °C leads to an apparent decrease of methanol STY and selectivity, which resulted from methane over-oxidation and methanol reforming (see the Supplementary Information). Increasing the Cu/Al ratio from 0.05 to 0.22 also results in an apparent decrease of the methanol STY (Fig. 2b, Supplementary Fig. 10) accompanied by the appearance of Cu-OH single sites and Cu-oxo clusters on 8MRs. Besides, temperature-programmed surface reactions (TPSR) show that the production of methanol over Cu/SSZ-13 catalysts is accompanied by the formation of H$_2$ and CO$_2$ (Supplementary Fig. 11), and the lower CO$_2$ yield of Cu$_1$/SSZ-13 reflects a weaker ability of methane over-oxidation compared with the other catalysts with higher Cu/Al ratios (see the details in Supplementary Information). Additionally, such a single Cu active site confined on a small 6MR void was also used for the aerobic MTM reaction, the high methanol yield and selectivity were also achieved at higher reaction temperatures (>350 °C), with a low

CO$_2$ production (see the details in Supporting Information, Supplementary Fig. 12).

## In situ FTIR spectroscopy

To further confirm that the Cu(II) single sites on 6MR are active sites in methane-steam to methanol reaction, in situ FTIR spectroscopic analysis of Cu$_1$/SSZ-13 was performed at different reaction temperatures (300-400 °C). As shown in Fig. 3a, the characteristic band of 6MR-hosted bare Cu(II) single sites is observed at ~900 cm$^{-1}$ in the flowing He. Introducing the reaction gas mixture (CH$_4$ + H$_2$O) to Cu$_1$/SSZ-13 at 400 °C resulted in an apparent band at ~2153 cm$^{-1}$, which is attributed to the formation of carbonyl group on Cu$^+$ centers[9], accompanied with a decrease of the band intensity at ~900 cm$^{-1}$ (Fig. 3c iii), suggesting that the Cu(II) single-atom site on 6MR is active in MTM conversion. The increase in relative intensity of the band appearing at ~950 cm$^{-1}$ (Fig. 3c ii) is ascribed to the formation of Cu-CH$_3$ intermediate species on 6MR, resulting from methane C-H activation on Cu atoms, based on DFT simulations (Supplementary Fig. 2). However, the variations of these bands during MTM become less apparent at lower temperatures (Fig. 3a). This reveals that high MTM conversions on 6MR-confined Cu single sites only occur at higher temperatures, which is crucial to trigger a Cu$^{2+}$-Cu$^+$ redox cycle, as evidenced by an apparent increase of the band intensity at 2153 cm$^{-1}$ with the elevated temperatures (Fig. 3c i). For the Cu/SSZ-13 with higher Cu/Al ratios, MTM conversions on 6MR-confined Cu single sites are also found via in situ FTIR spectroscopy (Supplementary Fig. 13). Such active sites on 6MRs for MTM conversions are also demonstrated by in situ DR UV-Vis spectroscopy (Supplementary Fig. 14). It is worth noting that the decrease of the band intensity at ~900 cm$^{-1}$ after feeding the reaction gas is ascribed to the C-H activation of methane on the Cu single-atom to form Cu-CH$_3$ intermediate species, instead of water activation, because the bands at ~900 cm$^{-1}$ did not change when Cu$_1$/SSZ-13 was only exposed in the flowing steam, while the bands appearing at ~950 and ~2153 cm$^{-1}$ were detected only in methane-containing reaction gas (Fig. 3b and c iv). Besides, the in situ FTIR spectra of Cu$_1$/SSZ-13 reveal that MTM is accompanied with an increase of Brønsted acid sites (Si-O(H)-Al, ~3585 cm$^{-1}$)[30] and a formation of Cu-OH species (~3655 cm$^{-1}$)[30,33] from the transformation of tetra-coordinated Cu(II) single-atoms located on 6MR (Fig. 3d).

## Isotope labeling of methane-steam reaction pathway

Isotope labeling experiments using CH$_4$-D$_2$O mixtures (Fig. 4a, Supplementary Fig. 15) indicate that the exchange of H atoms in CH$_4$ with D atoms in D$_2$O occurred on single Cu sites and/or Brønsted acid sites in Cu$_1$/SSZ-13 during the MTM reaction (see the details in Supporting Information). The formation of CH$_3$D at higher temperatures is due to a combination of the methyl formed from CH$_4$ activation and the D atom from D$_2$O dissociation, which may be an important reason for low methanol yield at high temperatures (Fig. 2) due to the thermodynamically-favored fortune of the methyl towards methane but not methanol. The production of HD reveals the combination of H atoms from methane and water in the continuous MTM process, accompanying with the CH$_3$OD formation (Fig. 4c). In contrast, the formation of H$_2$ is ascribed to the reaction of HDO and CH$_4$ (CH$_4$ + HDO → CH$_3$OD + H$_2$, Fig. 4b). High H$_2$ and HD production at high temperatures are attributed to the methanol reforming, which also lowers methanol yields. The production of CH$_3$OH at higher temperatures (>500 °C, Fig. 4c) is due to the presence of H$_2$O formed from H/D exchange (CH$_4$ + HDO → CH$_3$D + H$_2$O). Besides, the $^{18}$O-labeling experiments in CH$_4$-H$_2$$^{18}$O reaction systems confirm that water is the dominant oxidant in MTM despite the existence of a trace amount of ambient O$_2$ (Supplementary Fig. 16).

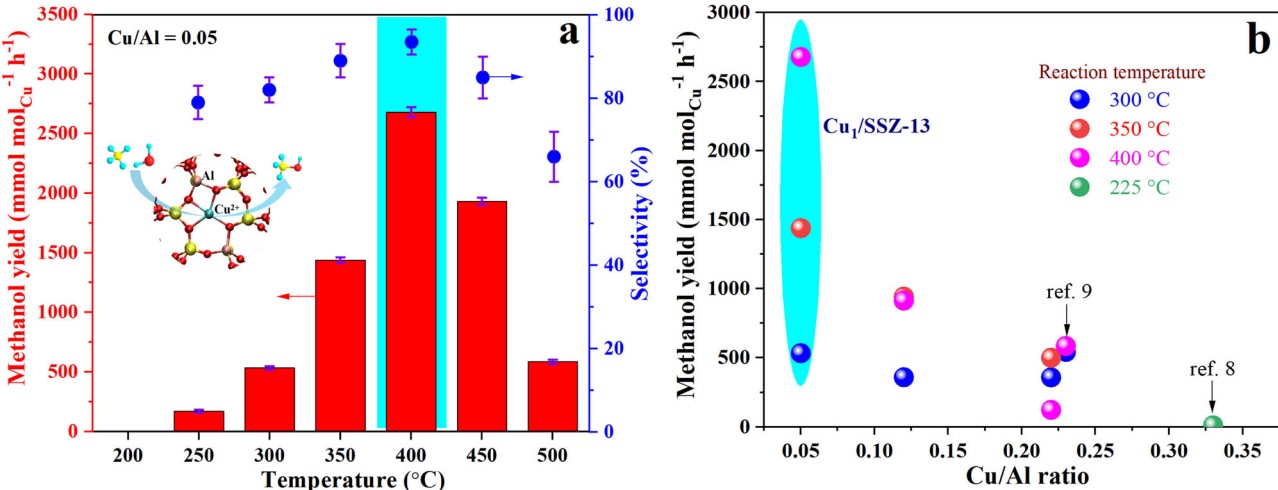

**Fig. 2 | Catalytic activity for methane-steam to methanol. a** Selective methane-steam conversion to methanol on $Cu_1$/SSZ-13 catalyst (Cu/Al = 0.05) under anaerobic condition upon elevating reaction temperatures from 200 to 500 °C. Red bars and blue points refer to methanol yield (mmol $mol_{Cu}^{-1}$ $h^{-1}$) and methanol selectivity (%), respectively. **b** A comparison in methanol yields from $Cu_1$/SSZ-13 (Cu/Al = 0.05) with only 6MR-confined Cu single sites and the other Cu/SSZ-13 catalysts (Cu/Al > 0.05) containing 8MR-hosted Cu-OH sites. Reaction conditions: 100 mg catalyst, total flow rate of 15 ml $min^{-1}$; 90% $CH_4$, 3.2% $H_2O$ and He balance. The optimal methanol yields highlighted with cyan color.

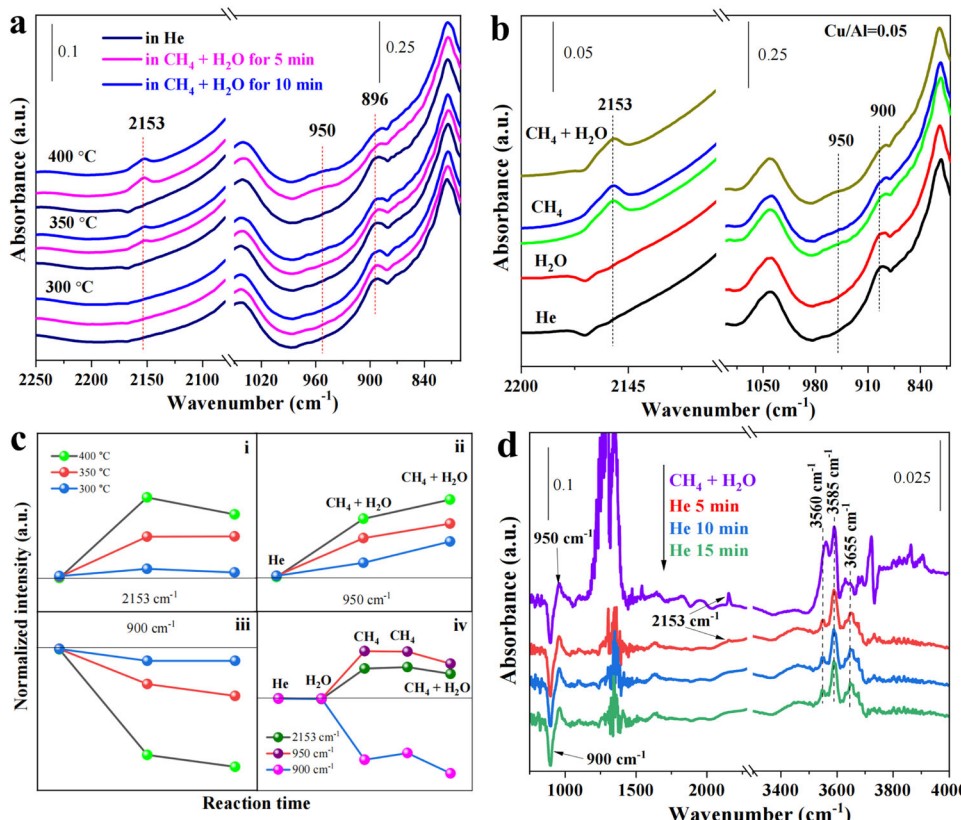

**Fig. 3 | in situ FTIR characterization. a** in situ FTIR spectra of $Cu_1$/SSZ-13 catalyst in MTM at different temperatures (300-400 °C). **b** Steady-state FTIR spectra of $Cu_1$/SSZ-13 catalyst under different atmospheres at 400 °C. **c** Changes in the intensities of the selected IR bands in (**a**) and (**b**). **d** in situ FTIR spectra of $Cu_1$/SSZ-13 during methane-steam conversion at 400 °C, the sample was firstly exposed in $CH_4 + H_2O$ and then in He. Conditions: ~30 mg catalyst, total flow rate of 15 ml $min^{-1}$; 90% $CH_4$, 3.2% $H_2O$, He balance.

## Density functional theory (DFT) analysis

Finally, a plausible pathway for MTM over the 6MR-confined Cu single site is proposed based on DFT calculations. As shown in Fig. 5a, the Cu single atom on the 6MR first transfers to monocopper Cu-OH site in the co-existence of methane and water with a low activation energy of 0.6 eV (Supplementary Fig. 17) based on the in situ FTIR results of the spent $Cu_1$/SSZ-13 mentioned above (Fig. 3d). In contrast, the direct conversion of methane on the bare tetra-coordinated Cu(II) single

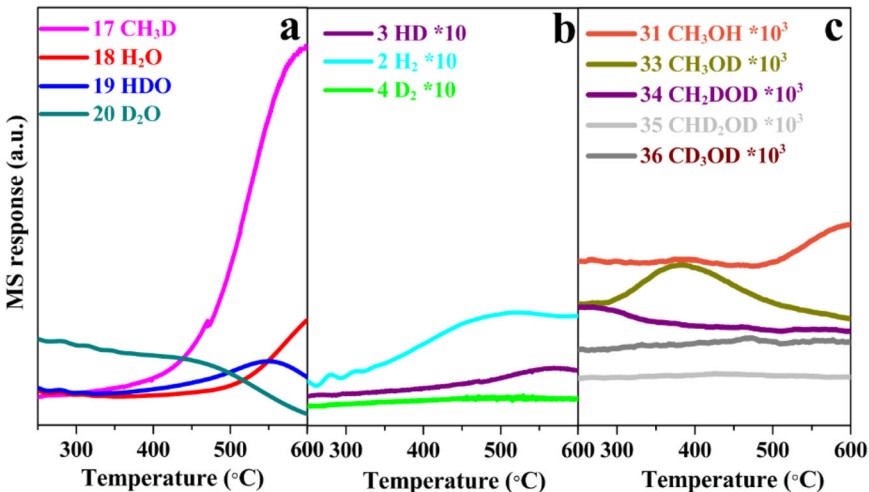

**Fig. 4 | Temperature-programmed surface reactions (TPSR) of continuous MTM on Cu₁/SSZ-13 catalyst with isotope labeling CH₄-D₂O system. a** Formation of CH₃D, H₂O, HDO, and D₂O. **b** Formation of HD, H₂, and D₂. **c** Formation of

CH₃OH, CH₃OD, CH₂DOD, CHD₂OD, and CD₃OD. Reaction conditions: 100 mg catalyst, total flow rate = 15 ml min⁻¹; 96.8% methane, 3.2% water.

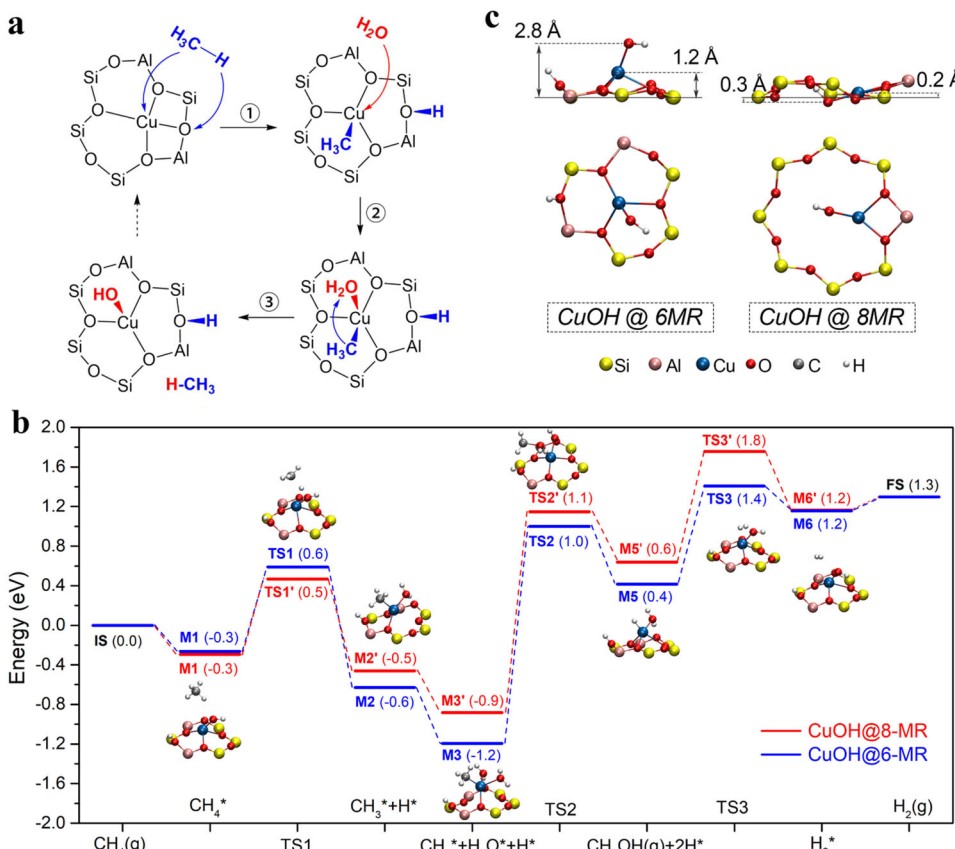

**Fig. 5 | Plausible pathway for MTM. a** The structure diagram of conversion from bare Cu(II) atom to Cu-OH single site on 6MR in the co-existence of methane and water. **b** The energy diagrams of methane-to-methanol using water as the oxidant on Cu-OH single sites from 6MR (blue line) and 8MR (red line). The optimized

ground-state structures of all intermediates (M1-M6) and transition states (TS1-TS3) were labeled around the pathway. **c** The distance from Cu atom and O atom to the ring plane for Cu-OH single sites on 6MR and 8MR.

atom has a much higher energy barrier of 1.7 eV (Supplementary Fig. 17). It is thus reasonable to propose the Cu-OH species that comes from the transformation of bare Cu(II) single-atoms as the active sites for MTM over Cu₁/SSZ-13. Figure 5b shows the pathway of methane-steam to methanol on the 6MR-hosted Cu-OH single sites. The C-H

activation of methane via H-abstraction by oxygen atom results in the evolution of Cu-OH into HOH*-Cu-CH₃* structure (**M2**) through the transition state **TS1** with an energy of 0.6 eV. On the other hand, the H-abstraction by a Cu atom from Cu-OH site is difficult due to higher activation energy to break the C-H bond (2.1 eV, Supplementary

Fig. 18). The subsequent adsorption of water on the Cu atom produces a HOH*-Cu(H₂O)-CH₃* complex (**M3**), which then forms a methanol molecule (**M4**) via a transition state (**TS2**) with an energy of 1.0 eV and next converted to HOH*-Cu-H* species (**M5**) by methanol desorption. Finally, the combination of the H* atoms from Cu-H* and HO-H* produced H₂ via transition state **TS3** (1.4 eV), accompanied by the regeneration of the Cu-OH active site (**M6**).

In this pathway, the H₂ formation from two H atoms is the rate limiting step (**M5 → M6**) as shown in Fig. 5b, which is affected by the ring framework where the Cu-OH active site located. In comparison, the energy of the dehydrogenation transition state (**TS3'**) for 8MR-hosted Cu-OH single sites (see the optimized structures in Supplementary Fig. 19) is 0.4 eV higher than that for 6MR-hosted ones. According to the structural analysis, the smaller diameter of 6MR forces the Cu-OH site away from the plane of the ring (Fig. 5c). As a result, the distances from the Cu and O atoms to the plane of the 6MR are 1.2 and 2.8 Å, respectively, which are much larger than the Cu-OH single site on 8MR (0.2 and 0.3 Å). The planar structure of 8MR-hosted Cu-OH sites is found to be unfavorable for the formation of the rate-limiting transition state, which causes a higher energy (0.43 eV) for the structural distortions of the framework than the case of 6MR (0.17 eV) (Fig. 6a). This effect of spatial confinement originated from the orientation of Cu-OH single sites in the 6MR and 8MR voids of SSZ-13 is

also true for several adsorbed intermediates involved in the MTM pathway (Fig. 6c). Based on this, the planar Cu-O-Cu sites on 8MR are also considered and show a higher energy barrier of 1.7 eV in H₂ formation than Cu-OH on 6MR (Supplementary Fig. 20). Therefore, the non-planar feature of the Cu-OH site residing on 6MR circumvents the spatial confinement by the small voids as found for 8MR, accounting for the uniquely high catalytic activity for the methane-steam conversion.

In this study, we demonstrate that Cu-OH single sites confined within the 6-membered ring (6MR) void of SSZ-13 zeolite exhibit high efficiency in continuous methane-to-methanol conversion using water as the oxidant. The optimized reaction conditions result in a high methanol space-time yield of 2678 mmol mol$_{Cu}^{-1}$ h$^{-1}$ and 93% selectivity at a temperature of 400 °C. On the other hand, the presence of 8MR-confined active sites (Cu-oxo clusters and Cu-OH sites) leads to over-oxidation of methane into CO₂ and reduces methanol space-time yield and selectivity. The high catalytic efficiency of 6MR voids is attributed to their stronger spatial confinement effect compared to the larger 8MR voids, based on theoretical calculations. This highlights the suitability of 6MR voids as hosts for Cu-OH single sites for improved methane-to-methanol conversions. The results also show that while bare Cu(II) single-atom sites hosted in 6MR are active in methane C-H activation, their stable tetra-coordinated structure

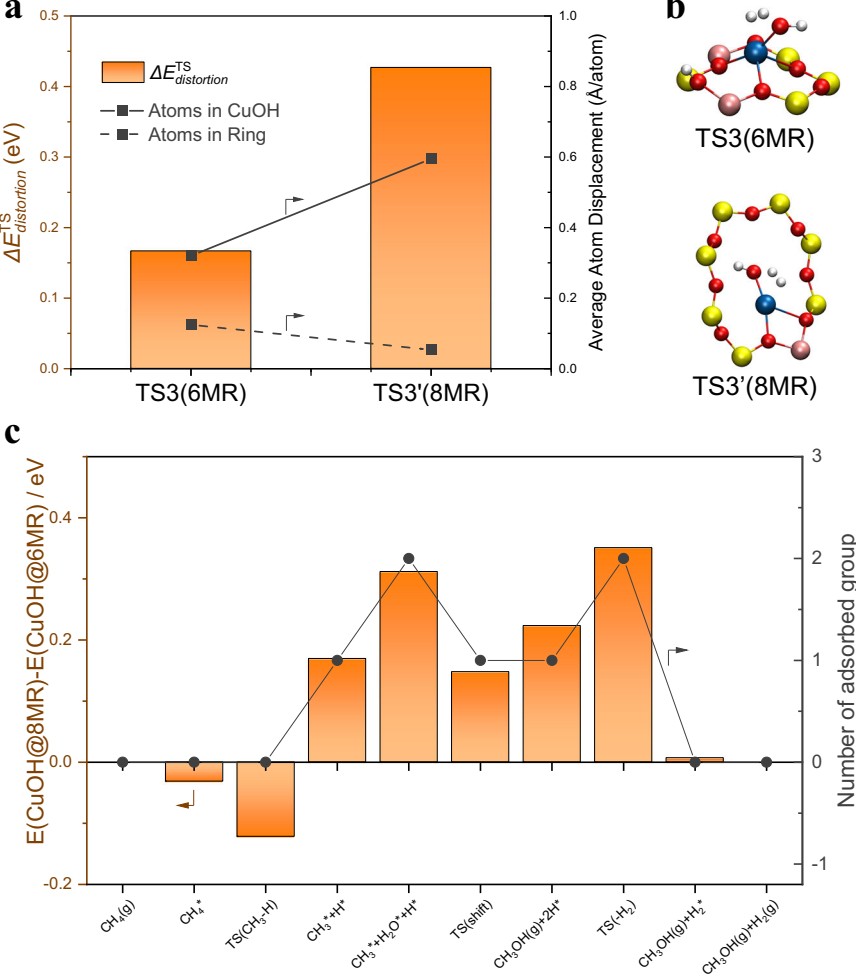

**Fig. 6 | Energy change and optimized transition-state structures. a** The column chart of the energy change of structural distortions for TS3 and TS3' in Fig. 5b. The line chart shows their average atom displacement of Cu-OH group (solid line) and the ring where Cu-OH site located (dotted line). **b** The optimized transition-state

structures for TS3 and TS3'. **c** The energy change of each structure and the number of adsorption groups on Cu-OH sites in the MTM reaction process as shown in Fig. 5b.

hinders their reactivity at lower temperatures (≤200 °C). A possible reaction pathway for the methane-to-methanol conversion on 6MR-confined Cu(II) single sites is proposed based on in situ spectroscopy, isotope labeling, and DFT calculations. In conclusion, this study provides valuable insights into the understanding of Cu-exchanged zeolite catalysts for continuous methane-steam to methanol conversion and paves the way for further advancements in this field.

## Methods

### Preparation of Cu-exchanged zeolites

Hydrogen form zeolite H-SSZ-13 (Si/Al = 10) was obtained by commercial purchase, its morphology feature was measured via scanning electron microscope (SEM) analyzer (GeminiSEM 500, Zeiss) and shows the shape of cube with the size of 1.5–2 μm (Supplementary Fig. 21). The copper-exchanged SSZ-13 zeolite samples with different Cu/Al ratios were prepared through the following procedure: 6 g of zeolite was firstly pretreated at 500 °C in air for 2 h to desorb the water adsorbed in the zeolite. Then 2 g of zeolite powder was stirred in 100 ml of copper(II) nitrate trihydrate (Cu(NO$_3$)$_2$·3H$_2$O, 99.99%, Sinopharm Chemical Reagent Co., Ltd) at 80 °C. The loading amount of copper and the Cu/Al ratios between 0.05 and 0.22 were determined by controlling the concentration of copper(II) nitrate solutions and the ion-exchange time. The pH of the solution was 5.0-5.8 during ion exchange. Subsequently, the suspension was centrifugalized and rinsed 3-4 times with deionized water (50 ml each time) to remove the surplus copper ions and ensure the pores in zeolites only contained the exchanged copper ions. The copper-exchanged samples were finally dried via a slow step-drying process at the temperatures ranging from 40 to 120 °C[34] and calcined in static air at 550 °C for 6 h with a heating rate of 1 °C min$^{-1}$. Copper (Cu), aluminum (Al) and silicon (Si) contents were measured by wavelength dispersive X-ray fluorescence (WDXRF) on a S8 TIGER X-ray fluorescence spectrometer (Bruker, Germany). The X-ray fluorescent spectra of Cu/SSZ-13 samples were collected under vacuum condition with the working voltage of 60 kV and the electric current of 40 mA.

### Characterization

X-ray diffraction (XRD) patterns of Cu-exchanged SSZ-13 samples were measured on Rigaku Ultima IV using Cu Kα radiation (λ = 0.15406 nm, 40 kV, 40 mA). The surface area and micropore volume of Cu-exchanged SSZ-13 were measured by N$_2$ adsorption-desorption at -196 °C on an automated analyzer (Micromeritics TriStar II). Prior to the measurement, the samples were degassed under vacuum at 300 °C for 5 h. The specific surface areas and the average pore diameters were obtained via Brunauer-Emmett-Teller (BET) and Barret-Joyner-Halenda (BJH) methods, respectively. High angle annular dark filed scanning transmission electron microscopy (HAADF-STEM) was carried out to confirm the presence of copper single-atoms located on Cu$_1$/SSZ-13 catalyst on a FEI Talos electron microscope. Temperature-programmed reduction with hydrogen (H$_2$-TPR) was performed using TCD as detector to identify the copper species confined in SSZ-13 zeolite. Prior to H$_2$-TPR, the Cu ion-exchanged SSZ-13 (100 mg) was pretreated at 400 °C in the flowing He (30 ml min$^{-1}$) for 30 min, then cooled down to room temperature (RT), H$_2$-TPR was carried out in the flowing 5% H$_2$/Ar (30 ml min$^{-1}$) at a ramp of 8 °C/min. The signal of H$_2$ consumption was recorded by TCD detector. FTIR spectra of Cu/SSZ-13 samples were measured on a Nicolet IS50 FTIR spectrometer in the diffuse reflectance mode equipped with a high sensitivity mercury-cadmium-telluride (MCT) detector and a high-temperature reaction cell with KBr window. The sample was first pretreated at 400 °C in the flowing He for 30 min, then the temperature was controlled to 200 °C and the IR spectrum of the sample was recorded using KBr powders as the background. Diffuse reflectance UV-Vis spectra of Cu/SSZ-13 samples were measured to identify copper species on a Cary Series UV-Vis

spectrophotometer (Agilent Technologies) in the diffuse reflectance mode equipped with a high-temperature reaction cell. The sample was firstly diluted with BaSO$_4$ powders at a mass ratio of 1/2 and pretreated in the flowing He (30 ml min$^{-1}$) at 400 °C for 30 min, then cooled down to RT and the spectrum was collected, using BaSO$_4$ as the background. In situ UV-Vis spectra of Cu/SSZ-13 catalysts in reaction gas (CH$_4$ + H$_2$O) were performed under the same conditions. Electron paramagnetic resonance (EPR) spectrum of Cu/SSZ-13 (Cu/Al = 0.05) was measured on X-band Bruker EMX EPR spectrometer in the range of 220-400 mT. The fine powder sample (-60 mg) was placed in a quartz EPR tube (I.D. = 4 mm). The EPR spectrum was collected at low temperature (−163 °C) by subtracting the background spectrum of an empty tube. X-ray absorption spectroscopy (XAS) measurements at the Cu K-edges were conducted at the CLAESS beamline, ALBA synchrotron (Spain) using a Si(311) monochromator with an incident energy resolution of 0.3 eV. The harmonic rejection was achieved by choosing a proper angle and coating of a collimating and focusing mirror. The measurements were carried out in the trans mission mode and Cu foils were used for energy calibration. X-ray absorption near edge structure (XANES) data were processed with the ATHENA software package. Extended x-ray absorption fine structure (EXAFS) data were processed and fitted with the ARTEMIS software package. Prior to XAS measurements, the powders were pressed into pellets.

### Testing of activity for continuous methane-steam to methanol conversion

Continuous methane-steam conversion to methanol was carried out in a tubular reactor (U-shaped quartz tube, I.D. 8.0 mm) with a continuous flow of 15 ml min$^{-1}$ (GHSV = 9000 ml g$_{cat.}$$^{-1}$ h$^{-1}$) under the pressure of 1 bar. The reactor was heated through a single-zone furnace. K-type thermocouple was mounted into the oven and used to control temperature. 100 mg of Cu/SSZ-13 sample (40-60 mesh, 0.3-0.45 mm) were packed between quartz wool plugs in the reactor and placed in the furnace heating zone. The reaction gas mixture, i.e., 90% CH$_4$, 3.2% H$_2$O, He as the balance, was fed into the reactor and their flow rates were controlled by independent mass flow meters (Bronkhorst). The steam was introduced into the reactor through a saturator maintained at 25 °C and used as the source of oxygen. Prior to the measurement, the deionized water in an airtight saturator was heated up to 80 °C and kept at this temperature for 24 h in the flowing helium to remove the potential O$_2$ impurity in water. However, there was still -90 ppm residual oxygen in the pretreated water (see Supplementary Fig. 22). In the continuous methane-steam conversion process, the Cu-exchanged SSZ-13 was firstly pretreated in the flowing helium (15 ml min$^{-1}$) at 400 °C for 30 min with a heating rate of 5 °C min$^{-1}$ and then exposed in the reaction gas mixture in a reaction temperature range of 200-500 °C. The effluent products were identified and quantified on a mass spectrometer (MS, ThermoStar, GSD320, Shanghai, China) based on MS signals of methanol (CH$_3$OH), dimethyl ether ((CH$_3$)$_2$O) and carbon dioxide (CO$_2$) at $m/z$ = 31, 45 and 44. The calibration curves of CH$_3$OH, CO$_2$ and (CH$_3$)$_2$O were obtained by an external standard method (see Supplementary Fig. 23). The space-time yields of the products were calculated as follows:

$$Y_i = \frac{k_i \times M_{Cu}}{m \times x\%} \int_0^{60} I_i dt$$

where $Y_i, k_i, I_i, M_{Cu}, m, x\%$ and $t$ refer to the space-time yield of $i$-product (mmol mol$_{Cu}$$^{-1}$ h$^{-1}$), the calibration coefficient, the MS intensity of $i$-product, the molar mass of copper (g mol$^{-1}$), the mass of Cu/SSZ-13 used for methane-steam conversion (0.1 g), the content (wt.%) of copper in per gram Cu/SSZ−13 catalyst and the continuous reaction time (min), respectively.

The selectivity of methanol ($S_{CH3OH}$) was calculated as the following equation:

$$S_{CH3OH} = \frac{Y_{CH3OH}}{\sum i Y_i}$$

### Temperature-programmed surface reaction (TPSR) coupled with isotope labeling

Temperature-programmed surface reaction were carried out in the fixed-bed reactor. 100 mg of Cu/SSZ-13 sample (40-60 mesh, 0.3-0.45 mm) was placed in a U-shape quartz reactor (I.D. 8.0 mm) and pretreated at 400 °C in He flow (15 ml min$^{-1}$) for 30 min. After the temperature dropped to 120 °C, the reaction gas mixtures including $CH_4$ (96.8%) + $H_2O$ (3.2%), $CH_4$ + $D_2O$ or $CH_4$ + $H_2^{18}O$ were fed to the reactor with the total flow rate of 15 ml min$^{-1}$. To avoid the impact of $O_2$ impurity in water, the deionized water was pretreated at 80 °C in the flowing helium for 24 h. TPSR coupled with deuterium and $^{18}O$ labeling experiments ($D_2O$ or $H_2^{18}O$ as reagents) were performed to reveal the possible methane-water conversion pathway. In the TPSR mode, the sample was heated from 120 to 650 °C at a rate of 8 °C min$^{-1}$. The reaction outlet composition was analyzed on-line on the MS (Pfeiffer Omnistar GSD320, Shanghai, China) based on the time-dependent evolution of m/z signals of the products (m/z values, see Supplementary Table 4).

### In situ FTIR spectroscopy

In situ FTIR spectroscopy experiments of Cu/SSZ-13 samples were performed in continuous reaction process on Nicolet IS50 FTIR spectrometer in the diffuse reflectance mode equipped with a MCT detector. The sample (~30 mg) was first pretreated at 400 °C in the flowing helium for 30 min, and then cooled down to the reaction temperatures (200-400 °C), the continuous reaction gas mixture (90% $CH_4$ + 3.2% $H_2O$ + He balance) was fed into the high-temperature reaction cell. Subsequently, the IR spectra of the samples were recorded by subtraction of the background spectrum of KBr or the catalyst itself at the reaction temperature. Besides, for insight into the real reaction case in the co-existence of $CH_4$ and $H_2O$, the pretreated Cu/SSZ-13 was respectively exposed in the flowing 3.2% $H_2O$/He and 90% $CH_4$/He reactant gases at the reaction temperature of 400 °C to investigate which one molecule ($H_2O$ or $CH_4$) was firstly activated on Cu active sites.

### Computational method and modeling

All periodic density functional theory (DFT) calculations were performed using the Vienna Ab initio Simulation Package (VASP) software[35–38] with a generalized gradient-approximated (GGA) Perdew-Burke-Ernzerhof (PBE) exchange-correlation functional[39]. A plane-wave basis with a cutoff energy of 400 eV was employed. The Brillouin zone was sampled using the gamma point, whereas the Gaussian smearing was set at 0.05 eV. Sin-polarized calculations were performed throughout this study. The semiempirical Grimme's D3 correction was employed to include van der Waals (vdW) interactions[40]. The transition states were determined by the climbing-image nudged elastic band method (CI-NEB)[41,42] and optimized using dimer method[43]. The energy and force convergence criteria for all structures were set to 10$^{-7}$ eV and 0.05 eV Å$^{-1}$, respectively.

The model for Cu$_1$/SSZ-13 zeolite with 6MR-hosted mononuclear bare copper was built as the initial state of active site (see the Supplementary Fig. 24). The all-silica CHA-type zeolite framework (a = b = 13.675 Å, c = 14.767 Å, and γ = 120°) was used as the initial model. The -Si-O-Si-O-Si- in the 6-membered (6-MR) ring was replaced by -Al-O-Si-O-Al- to combine the copper cation[44]. One single Cu atom was located inside the 6-MR ring. One Si atom was fixed to avoid the translation and rotation of the zeolite framework during optimization, while all the remaining atoms were allowed to relax.

The energy changes of structural distortions of the transition states TS3 and TS3' ($\Delta E^{TS}_{distortion}$) are used to account for the effect of the adsorbed species on the Cu-OH site and zeolite framework, which are calculated as follows[45]:

$$\Delta E^{TS}_{distortion} = E_{TS-2H} - E_{CuOH/CHA}$$

where $E_{TS-2H}$ refers to the total electronic energy of the rest part of the transition state model after removing the adsorbed species, which are two hydrogen atoms in TS3 and TS3'. $E_{CuOH/CHA}$ is the total electronic energy of a given Cu-OH-containing CHA model.

IR spectrum simulation was assessed by using the PHONOPY code (version: 2.12.0)[46] with forces resulting and Born charge from VASP calculations and processing the results with python scripts from the method of Janine George, et al. (version: 1.0.4)[47–50].

## Data availability

All data needed to evaluate the conclusions in the paper are present in the paper and/or the Supplementary Information. The raw data sets used for the presented analysis within the current study are available from the corresponding authors on request.

## Code availability

The code that support the findings of this study is available from the corresponding author upon request.

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

## Acknowledgements

Funds from the National High-Level Young Talents Program and National Natural Science Foundation of China (22072118, 22121001 and) are acknowledged. We also thank the financial support from State Key Laboratory for Physical Chemistry of Solid Surfaces of Xiamen University. Part fund was supported by Science and Technology Projects of Innovation Laboratory for Sciences and Technologies of Energy Materials of Fujian Province (IKKEM) (HRTP-[2022]-3) and the Fundamental Research Funds for the Central Universities (20720220008).

## Author contributions

H.Z. and P.H. contributed equally to this work. H.Z., Y.W. and H.X. conceived the presented idea. H.Z. carried out the experiments. P.H. and S.W. performed DFT calculations. C.D. conducted HAADF-STEM experiments and analysis. H.Z., P.H., J.Z. and D.W. analyzed and visualized the data. H.Z. and P.H. wrote the manuscript with input from all authors. H.X. and S.W. provided supervision and resources for this work.

## Competing interests

The authors declare no competing interest.
