## [Peer Review File · Nature Communications]

Reviewers' Comments:

Reviewer #1:

Remarks to the Author:

In this manuscript, the authors intensively investigated the direct selective oxidation of methane into methanol over Cu-SSZ-13 under anaerobic conditions. This is a well-organized manuscript. However, the following comments should be carefully considered during the revision process.

- 1) The activity test should be done very carefully in the case of aerobic conditions. Since only small amounts of oxygen in the feed can significantly increase the methanol productivity in this case, the oxygen concentration should be checked for each activity test. Did you check the oxygen concentration?
- 2) It is highly plausible to carry out the labelling experiment with H₂¹⁶O to exclude any possibility of ambient O₂ participation.
- 3) It is highly recommended to present the equilibrium conversions of methane under reaction conditions adopted in this manuscript. The maximum methanol productivity can also be derived and compared with those reported in this work.
- 4) Unfortunately, the authors compared single Cu-oxo species in 6-MR ring with that in 8-MR ring with DFT calculation. As you know, the latter one is known to be less active than the dimeric Cu-oxo species. It would be better to compare single Cu-oxo species in 6-MR ring with other Cu-oxo species including dimeric and trimeric Cu-oxo species additionally.

Reviewer #3:

Remarks to the Author:

Reviewer's comments to "Breaking the barrier: New insights into the direct oxidation of methane to methanol (MTM) with confined Cu-OH single sites in SSZ-13 zeolite" by Zhang et al.

Zhang et al have reported their work on direct conversion of methane to methanol with water as oxidant over the SSZ-13 zeolite. Cu-OH is proposed to be the catalytic centre and a large fraction of the work is dedicated to this topic. Methane activation (i.e. controlled selective C-H bond activation) has been one of a set of important chemical reactions that is deemed to be important from an industrial and a scientific point of view. C-H activation has most commonly been demonstrated in homogeneous catalytic systems using transition metals like e.g. Pt. Over the last 10-15 years there has been an increased interest in heterogenized catalysts for methane oxidation to methanol. Most works have reported a stepwise path for achieving the goal, but lately some reports have shown catalytic conversion of methane over Cu-zeolites using oxidants. The selectivity, and especially the conversion, were rather modest. Zhang et al. takes the Cu catalyzed CH activation a major step further with their good selectivity and conversion (a high space time yield).

The work is presented very well and convincingly and is of high quality, however, there are some issues that should be considered/discussed in the article before publication.

In general, it appears that the reaction might take place as suggested, but most data give only circumstantial evidence. A reaction mechanism proving catalytic conversion of methane with a co-reactant along a specific path, and not instead some other path, or even sequential reaction is hard to prove. In fact, if formulated as a hypothesis you can never prove it, only disprove it. That leads to my point, I think there is a lack of evidence for what mechanisms that do not take place. This sounds strange but as the text is written all data except conversion and selectivity is focused on arguing for an apparently chosen mechanism, and not on what mechanism that is going. This is a subtle difference here. There are a lot of arguments for the suggested mechanism, but there are e.g. no kinetic data giving the order of the reaction and no strong arguments that disprove other mechanisms.

- 1) Isotopic labelling. There should be a reference experiment using the same material except without Cu. The reason is that there can be exchange of H by CH₄ coordinating to Brønsted acid sites. A reference experiment at proper conditions would tell if H-D exchange on other sites than the suggested Cu-site is possible or not. For benzene such a reaction takes place, and it might also take place with methane.
- 2) Is the Cu-OH species shown by IR spectators or could there be other peaks from the really active catalytic sites? One could e.g. carry out an experiment where the catalyst is dried and

activated and methane reacted over it. It would be a small conversion. The next could be to pretreat the catalyst with water to make CuOH groups. One could then react methane over the CuOH material and evaluate the reactivity of the CuOH species.

3) The DFT looks nice, but there are rather high activation barriers, are there other possibilities that could be compared with? Own computations or others? What about coordination of more molecules like e.g. water? The calculated reaction path do not really prove anything without being discussed in a broader context.

Figure 3,

a) There are two different lines labelled CH₄+H₂O. What does that mean?, d) That part is hard to read and assess.

An important reference that is lacking is the one by Pappas et al. JACS 2017, 139, page 14961.

Reply to the comments of the Reviewers

#Reviewer 1

In this manuscript, the authors intensively investigated the direct selective oxidation of methane into methanol over Cu-SSZ-13 under anaerobic conditions. This is a well-organized manuscript. However, the following comments should be carefully considered during the revision process.

1) The activity test should be done very carefully in the case of aerobic conditions. Since only small amounts of oxygen in the feed can significantly increase the methanol productivity in this case, the oxygen concentration should be checked for each activity test. Did you check the oxygen concentration?

Response: Thank the reviewer very much for the suggestions. According to the reviewer's comments, we have performed the activity tests in the case of aerobic conditions and we have checked the oxygen concentration in our experiments by a MS, the activity results are shown in Fig. R1. It is revealed that the methanol productivity has an apparent increase with the increase of O₂ concentration at temperature of > 350 °C, but there is no visible increase at the temperature of ≤ 350 °C with the O₂ concentration. Furthermore, it was found that increasing the O₂ concentration to 500-1000 ppm resulted in a much higher yield of methanol at 400-500 °C (~4-4.5 mol mol_{Cu}⁻¹ h⁻¹), presenting a higher methanol selectivity (~93%). However, the small amount of oxygen in the feed (< 200 ppm) did not lead to a significant increase in the methanol yields at the same temperatures. Such a high methanol yield under 500-1000 ppm O₂ should be attributed to the role of oxygen in selective-oxidation of methane to methanol and/or the cooperative effect of O₂ and H₂O (Chem, 2021, 7, 1557-1568).

Revisions: We have added Fig. R1 above to the revised Supporting Information as Fig. S12. In the Page 8 and Page S23, we have added the above discussion into the revised manuscript and Supporting Information as highlighted in yellow.

Fig. R1. Selective methane-steam conversion to methanol on Cu₁/SSZ-13 catalyst (Cu/Al = 0.05) under aerobic condition upon elevating reaction temperatures from 300 to 500 °C and increasing oxygen concentration from 100 to 1000 ppm. Red cycles and blue squares refer to methanol yield (mol mol_{Cu}⁻¹ h⁻¹) and methanol selectivity (%), respectively.

2) It is highly plausible to carry out the labelling experiment with H₂¹⁸O to exclude any possibility of ambient O₂ participation.

Response: We thank the reviewer for the suggestion. Based on the reviewer's comment, we have carried out the labeling experiments with H₂¹⁸O, the results are shown in the Fig. R2 below, which has been included as Fig. S16 in the revised Supporting Information. In Fig. R2A, it was found that the MS signals of O₂ decrease at > 300 °C, this indicates that a trace amount of ambient O₂ participated in the chemical reactions. However, we did not find the apparent MS signals of CH₃OH and CO₂, except for the visible observation of CH₃¹⁸OH and C¹⁸O₂ signals, together with the increase of H₂O and the decrease of H₂¹⁸O. The similar results were also found in the aerobic reaction system of CH₄ + H₂¹⁸O + 100 ppm O₂. These results illustrate that water acted as the dominant oxidant in methane conversion.

Fig. R2. Temperature-programmed surface reactions (TPSR) of continuous MTM on Cu₁/SSZ-13 catalyst with isotope labeling CH₄ + H₂¹⁸O (A) and CH₄ + H₂¹⁸O + 100 ppm O₂ (B) systems. Reaction conditions: 100 mg catalyst, total flow rate = 15 ml min⁻¹; 96.8% CH₄, 3.2% H₂¹⁸O, 100 ppm O₂.

Similarly, in the work reported by L.D. Li, et al. (Chem, 2021, 7, 1557-1568), the O¹⁸-labeling experiments of MTM reaction were carried out under 400 ppm O₂, the results are copied and shown in the following Fig. R3. It was revealed that CH₃¹⁸OH produced from the reaction of CH₄ + H₂¹⁸O is the main product together with a small amount of CH₃OH (see Fig. R3A). Using ¹⁸O to label O₂ (Fig. R3B), the result also illustrates that the main MTM reaction is CH₄ + H₂O → CH₃OH, and water is the main oxidant in methane conversion. Meanwhile, the apparent O₂ consumption is also observed, this indicates that O₂ participated in the reaction. The authors thought that a stoichiometric ratio of H₂ and CH₃OH should be 1 if only H₂O is the oxidant, but they found the ratio is < 1, thus they speculated that O₂ participated in MTM reaction: CH₄ + H₂O + [O] → CH₃OH + H₂O.

Fig. R3. Temperature-programmed surface reactions of methane oxidation on Cu-CHA with isotope labeling $\text{CH}_4\text{-H}_2^{18}\text{O-O}_2$ system (A) and $\text{CH}_4\text{-H}_2\text{O-}^{18}\text{O}_2$ system (B). Reaction conditions: 0.1 g catalyst, total flow rate = 60 ml/min; 98% methane, 2% water, 400 ± 50 ppm dioxygen. Copied from the publication Chem, 2021, 7, 1557-1568 reported by L.D. Li, et al.

In our opinion, the O_2 participated in the reaction, but the oxygen source of methanol is only water, that is, the O_2 would not directly participate in MTM reaction. In the labeling experiments, we found that H_2^{18}O was consumed together with the production of H_2O , it is possible that H_2 produced from the reaction of CH_4 and H_2^{18}O can react with O_2 to form H_2O : $\text{CH}_4 + \text{H}_2^{18}\text{O} \rightarrow \text{CH}_3^{18}\text{OH} + \text{H}_2$, $\text{H}_2 + \text{O}_2 \rightarrow \text{H}_2\text{O}$ (overall reaction: $2\text{CH}_4 + 2\text{H}_2^{18}\text{O} + \text{O}_2 \rightarrow 2\text{CH}_3^{18}\text{OH} + 2\text{H}_2\text{O}$). Then the H_2O from H_2 oxidation participated in the MTM reaction, thus a trace amount of CH_3OH was found. In fact, the fast consumption of H_2 as the MTM product can enhance the production of CH_3OH .

Revisions: The above Fig. R2 and the discussion have been added in the revised manuscript (Page 11) and Supporting Information (Page S11, Page S25-26).

3) It is highly recommended to present the equilibrium conversions of methane under reaction conditions adopted in this manuscript. The maximum methanol productivity can also be derived and compared with those reported in this work.

Response: Thanks for the reviewer's comments. Based on the reaction: $\text{CH}_4 + \text{H}_2\text{O} \rightarrow \text{CH}_3\text{OH} + \text{H}_2$, we calculated the equilibrium conversions of methane under the

present reaction conditions. The equilibrium constant of 7.56×10^{-10} was obtained at the reaction temperature of 400 °C ($\Delta G = \sim 28.1$ kcal/mol), leading to a low methane-steam equilibrium conversion, with methanol yield of ~ 2.4 $\mu\text{mol}/\text{g}_{\text{cat.}}/\text{h}$. This value is lower compared with that reported in this work (~ 218 $\mu\text{mol}/\text{g}_{\text{cat.}}/\text{h}$). Based on the isotope labeling experiments with temperature-programmed surface reactions (TPSR) and *in situ* FTIR spectroscopy, the difference is explained as follows: 1) the H_2 produced from methane-steam reactions reacting with the ambient O_2 to produce H_2O ($\text{H}_2 + \text{O}_2 \rightarrow \text{H}_2\text{O}$) leads to the consumption of H_2 as the key product in MTM, which makes the equilibrium conversion shift toward a higher value, this is very favorable for higher methanol yields (overall reaction: $2\text{CH}_4 + 2\text{H}_2\text{O} + \text{O}_2 \rightarrow 2\text{CH}_3\text{OH} + 2\text{H}_2\text{O}$); 2) *in situ* FTIR experiments demonstrate the formation of B acid sites (H_B , ~ 3685 cm^{-1}) on 6MR during methane-steam reactions. B acid sites may participate in the reaction of MTM, which are reported by some references (Catal. Sci. Technol., 2018, 8, 4141-4150; Science 356, 2017, 523-527; Chem 7, 2021, 1557-1568). Therefore, the MTM conversion is not a simple reaction (i.e., $\text{CH}_4 + \text{H}_2\text{O} \rightarrow \text{CH}_3\text{OH} + \text{H}_2$) in Cu-zeolites, it may be: $2\text{CH}_4 + 2\text{H}_2\text{O} + \text{H}_\text{B} + \text{O}_2 \rightarrow 2\text{CH}_3\text{OH} + 2\text{H}_2\text{O} + \text{H}_\text{B}$. The two factors above resulted in the increase of the actual equilibrium conversion and a high methane conversion.

4) Unfortunately, the authors compared single Cu-oxo species in 6-MR ring with that in 8-MR ring with DFT calculation. As you know, the latter one is known to be less active than the dimeric Cu-oxo species. It would be better to compare single Cu-oxo species in 6-MR ring with other Cu-oxo species including dimeric and trimeric Cu-oxo species additionally.

Response: We thank the reviewer for the valuable suggestion. We have calculated the reaction pathway using the dimeric Cu cluster (Cu-O-Cu) accordingly as shown in Fig. R4. It is found that H_2 formation is the rate-limiting step with the highest energy barrier along the MTM reaction pathway. The H_2 formation (**TS3**) on Cu-O-Cu shows an energy barrier of 1.7 eV (Fig. R4), significantly higher than that on the 6MR hosted Cu-OH (1.4 eV, Fig. 5b in the main text). The higher activation energy for the dimeric Cu cluster is likely contributable to the planar structure of 8MR-hosted

Cu-O-Cu sites, which make the Cu sites unfavorable for the adsorption of two H atoms at the rate-limiting step.

Revisions: We have included Fig. R4 in Supporting Information as Fig. S20 and the above discussion has been incorporated in the revised manuscript (highlighted in yellow, Page 14 in the main text).

Fig. R4. The energy diagram of methane-to-methanol using water as oxidant on dimeric Cu cluster (Cu-O-Cu) from 8MR of Cu/SSZ-13 catalyst. The optimized ground-state structures of all intermediates (M1-M5) and transition states (TS1-TS3) were labeled around the pathway. The right showing the planar structure of Cu-O-Cu sites on 8MR.

#Reviewer 2

Reviewer's comments to "Breaking the barrier: New insights into the direct oxidation of methane to methanol (MTM) with confined Cu-OH single sites in SSZ-13 zeolite" by Zhang et al.

Zhang et al have reported their work on direct conversion of methane to methanol with water as oxidant over the SSZ-13 zeolite. Cu-OH is proposed to be the catalytic centre and a large fraction of the work is dedicated to this topic. Methane activation (i.e. controlled selective C-H bond activation) has been one of a set of important chemical reactions that is deemed to be important from an industrial and a scientific point of

view. C-H activation has most commonly been demonstrated in homogeneous catalytic systems using transition metals like e.g. Pt. Over the last 10-15 years there has been an increased interest in heterogenized catalysts for methane oxidation to methanol. Most works have reported a stepwise path for achieving the goal, but lately some reports have shown catalytic conversion of methane over Cu-zeolites using oxidants. The selectivity, and especially the conversion, were rather modest. Zhang et al. takes the Cu catalyzed CH activation a major step further with their good selectivity and conversion (a high space time yield).

The work is presented very well and convincingly and is of high quality, however, there are some issues that should be considered/discussed in the article before publication.

In general, it appears that the reaction might take place as suggested, but most data give only circumstantial evidence. A reaction mechanism proving catalytic conversion of methane with a co-reactant along a specific path, and not instead some other path, or even sequential reaction is hard to prove. In fact, if formulated as a hypothesis you can never prove it, only disprove it. That leads to my point, I think there is a lack of evidence for what mechanisms that do not take place. This sounds strange but as the text is written all data except conversion and selectivity is focused on arguing for an apparently chosen mechanism, and not on what mechanism that is going. This is a subtle difference here. There are a lot of arguments for the suggested mechanism, but there are e.g. no kinetic data giving the order of the reaction and no strong arguments that disprove other mechanisms.

Response: We thank the reviewer for the valuable comments. We agree with the reviewer that a reaction mechanism proving catalytic conversion of methane with a co-reactant along a specific path, and not instead some other paths, or even sequential reaction is hard to prove. In the present study, we proposed a reaction mechanism based on the experimental data including *in situ* FTIR spectroscopy, which is also confirmed by the D-/O¹⁸-isotope labeling tests appended in this revised version.

Firstly, we confirmed that a bare single Cu site was confined on the 6MR of SSZ-13 zeolite with a tetra-coordinated structure via a series of characterizations (e.g., FTIR,

UV-Vis, STEM, H₂-TPR, EPR and EXAFS), there is no other Cu species existing in Cu₁/SSZ-13. Therefore, the active sites for MTM were the tetra-coordinated single Cu site on 6MR. But DFT calculation shows a very higher energy (~1.7 eV) in methanol and H₂ formation (see Fig. S17). Then *in situ* FTIR spectroscopy revealed the formation of Cu-OH site as the active species in methane-water reactions (see Fig. 3d in the main text), which deriving from the transformation of the tetra-coordinated single Cu site on 6MR (see Fig. 5a in the main text). Then, the transformation of Cu site was investigated. By *in situ* FTIR experiments, it was found that exposing the single Cu site in 3.2% H₂O/He at 400 °C did not trigger its transformation to Cu-OH, the hydrolysis of bare single Cu sites on 6MR is difficult to occur even at 400 °C. While introducing CH₄ + H₂O to the single Cu sites triggered their transformation to Cu-OH (see Fig. 3 in the main text). This indicates that continuous methane-steam reaction led to the formation of Cu-OH. Taking account of the isotope labeling experiments, it was found that the methane molecules re-formed by the combination of a methyl from methane activation and a H from water dissociation, which is confirmed to occur on both Cu sites and B acid sites (see Fig. 4a in the main text and Fig. S15). According to these experimental data, we proposed a reaction pathway for the formation of Cu-OH active sites by DFT calculations, with a lower energy barrier (0.6 eV, see Fig. S17). Secondly, we proposed a reaction mechanism for methane-steam to methanol. The O¹⁸-labeling experiment that has been added in the revised manuscript demonstrated that water is the oxidant for methane-to-methanol reaction (see Fig. S16). In the reaction of MTM, it involves methane C-H activation, methanol formation and H₂ desorption. We proposed and compared two reaction pathways in which methane C-H bond activation respectively took place on Cu and O atoms in Cu-OH sites (see Fig. 5b and Fig. S18) via DFT. It is revealed that the pathway on Cu atoms needs higher energy to break C-H bond (2.1 eV) compared with that on O atoms (0.6 eV). Thus, we suggested that the C-H activation occurred on the -OH site of Cu-OH. Besides, the analysis of the reaction products based on the isotope labeling experiments shows that the -OH in CH₃OH mainly came from H₂O and the product H₂ from the combination of the H atoms in CH₄ and H₂O. These are important information in DFT study on the reaction mechanism. We found that the H₂

formation is the rate limiting step and the spatial confinement from 6MR is an important factor affecting the energy of the rate limiting step. Based on the above discussion, the reaction mechanism proposed in this work is reasonable.

To address the reviewer's concern, we have modified some sentences/paragraphs in the revised manuscript and Supporting Information and to show that the present work is also compromised to other possible mechanisms.

1) Isotopic labelling. There should be a reference experiment using the same material except without Cu. The reason is that there can be exchange of H by CH₄ coordinating to Brønsted acid sites. A reference experiment at proper conditions would tell if H-D exchange on other sites than the suggested Cu-site is possible or not. For benzene such a reaction take place, and it might also take place with methane.

Response: We thank the reviewer for the valuable comments. Based on the reviewer's suggestions, we have carried out a reference experiment using the same SSZ-13 zeolite without Cu, the results are shown in the following Fig. R1b (it has been included as Fig. S15 in the revised Supporting Information). It was found that the H from CH₄ molecule was exchanged with the H from H₂O, which is similar with the result shown in Fig. R2 (shown as Fig. 4 in the main text), but only occurring at the temperatures of > 450 °C. However, such results could not be found in a reference experiment without any materials (Fig. R1a). This illustrates that the exchange of H atoms between CH₄ and H₂O occurred on the surface of SSZ-13 zeolite, which, as mentioned by the reviewer, is attributed to the coordination of CH₄/H₂O to Brønsted acid sites. On the other hand, Fig. R2a shows that the H atom exchanges between CH₄ and H₂O over the Cu₁/SSZ-13 catalyst started to occur at lower temperature (~350 °C). This indicates that the H exchange also took place in the Cu sites under the present reaction conditions. We also investigated the H-exchange process occurred on the Cu single sites by DFT calculations, as shown in the following Fig. R3 (shown as Fig. S17). The activation of methane resulted in the evolution of tetra-coordinated copper located on 6MR into a Cu-CH₃* structure (**M2**), accompanied by the formation of Brønsted acid site (O_BH) on 6MR with energy barrier of 0.7 eV. The subsequent

adsorption of water produced a Cu complex ($\text{H}_2\text{O}^*\text{-Cu-CH}_3^*$, **M3**), which then re-formed a methane molecule via the methyl on Cu-CH_3^* abstracting a H from water dissociation, accompanied with the formation of monocopper Cu-OH species as the potential active sites (red line, **M3'** \rightarrow **M4'**). This pathway shows that a lower energy barrier of 0.6 eV (**TS2'**) is required in comparison with **TS2** (1.7 eV) on the blue line. Thus, it is rational to propose that the pathway of H exchanges also occurred on Cu sites.

In addition, Fig. R1b shows that the H exchange process led to the formation of CH_3D via the combination of CH_3 from CH_4 and D from D_2O , but we did not find the formation of CH_3OD . In comparison with the results in Fig. R2, it reveals that the production of CH_3OH only occurred on Cu sites hosted on 6MR. In fact, the methyl produced is unstable, and is easily combined with another H atoms to re-form the stable CH_4 molecule. On the contrary, it is more stable on the Cu sites by forming Cu-CH_3 species. Thus, the presence of Cu sites is crucial for the selective conversion of methane to methanol.

Revisions: The above discussion has been added in the revised manuscript and Supporting Information (Page 11, Fig. S15, Pages 25-26).

Fig. R1. The reference experiments of temperature-programmed surface reactions (TPSR) with

isotope labeling $\text{CH}_4\text{-D}_2\text{O}$ system, a: without any materials, only including the mixture of $\text{CH}_4 + \text{D}_2\text{O} + \text{He}$; b: with the zeolite of SSZ-13 without Cu site. Reaction conditions: 100 mg catalyst, total flow rate = 15 ml min^{-1} ; 96.8% methane, 3.2% water.

Fig. R2. Temperature-programmed surface reactions (TPSR) of continuous MTM on $\text{Cu}_1/\text{SSZ-13}$ catalyst with isotope labeling $\text{CH}_4\text{-D}_2\text{O}$ system. Reaction conditions: 100 mg catalyst, total flow rate = 15 ml min^{-1} ; 96.8% methane, 3.2% water.

Fig. R3. The energy diagram of methane-to-methanol using water as oxidant on isolated Cu single-atom site from 6MR of $\text{Cu}_1/\text{SSZ-13}$ catalyst (blue line), including the formation of monocopper Cu(II)-OH species (red line). The optimized ground-state structures of all intermediates (M1-M5) and transition states (TS1-TS3) were labeled around the pathway.

2) Is the Cu-OH species shown by IR spectators or could there be other peaks from the really active catalytic sites? One could e.g. carry out an experiment where the catalyst is dried and activated and methane reacted over it. It would be a small conversion. The next could be to pretreat the catalyst with water to make CuOH groups. One could then react methane over the CuOH material and evaluate the reactivity of the CuOH species.

Response: Thank the reviewer for the comments. Yes, the Cu-OH species can be shown by IR spectators. For the identification of the Cu-OH species, in this study, the *in-situ* IR spectroscopy of Cu₁/SSZ-13 was performed under the reaction gas mixture of CH₄ + H₂O at 400 °C, the IR spectra were recorded by subtraction of the background spectrum of the Cu₁/SSZ-13 catalyst itself at the reaction temperature (the background spectrum was collected in He), the results were shown in the following Fig. R4 (shown as Fig. 3d in the main text). It was found that, except for the apparent IR bands that are assigned to CH₄, a negative band at ~900 cm⁻¹ (as the characteristic band of bare tetra-coordinated Cu confined on 6MR) and a positive band at ~950 cm⁻¹ (as the characteristic band of Cu-CH₃) appeared after introducing the reaction gas of CH₄ + H₂O, illustrating the consumption of tetra-coordinated Cu single-atom sites and the formation of Cu-CH₃ species on 6MR during methane-steam conversion. Meanwhile, the Cu⁺-CO species and Brønsted acid sites were also detected at ~2153 and ~3585 cm⁻¹, respectively. This reveals that a Cu²⁺-Cu⁺ cycle occurred and the amount of B acid sites on 6MR increased during MTM. The Cu-OH species in Cu-zeolites is often identified by the characteristic band at ~3655 cm⁻¹ (Chem, 2021, 7, 1557-1568; ACS Catal., 2015, 5, 6780-6791), which could not be visibly observed in methane-steam reaction (violet color in Fig. R4). However, the characteristic band of Cu-OH was apparently detected in the spent Cu₁/SSZ-13 when it was exposed in the flowing He (other colors). Meanwhile, a negative band at ~900 cm⁻¹ did not disappear after methane reaction, this illustrates that the Cu-OH sites deriving from the tetra-coordinated Cu single-atom site on 6MR are the active sites in MTM. Besides, based on the experimental results shown in this work, we proposed a pathway to produce Cu-OH sites in the methane-steam reaction by DFT study, as

shown in the previous Fig. R3. It is revealed that forming Cu-OH deriving from 6MR-confined Cu single-atom site is facile via CH₄ activation and H₂O adsorption on Cu site due to a very low energy. Therefore, we proposed that the Cu-OH site is the active site for continuous MTM.

Fig. R4. *In situ* FTIR spectra of Cu₁/SSZ-13 during methane-steam conversion at 400 °C, the right showing the details in the range of 3300-4000 cm⁻¹. Reaction conditions: ~30 mg catalyst, total flow rate of 15 ml min⁻¹; 90% CH₄, 3.2% H₂O, He balance. After the reaction, the sample was exposed in the flowing He, then the IR spectra were recorded.

As suggested by the reviewer, we have compared the reactivities of the bare Cu and Cu-OH sites on 6MR. We first carried out the methane reaction experiment on Cu₁/SSZ-13 via introducing 90%CH₄/10%He (without water) to the dry catalyst at 300 °C, no MS signals of CH₃OH was detected. This indicates that the absence of water cannot induce the formation of CH₃OH due to the absence of water. In the present study, the result of *in situ* FTIR spectroscopy (Fig. 3b in the main text) indicates that the activation of CH₄ on isolated Cu sites resulted in the formation of Cu-CH₃ sites. In the activity measurement, after methane activation on the isolated Cu sites, the catalyst was purged by the flowing He, then the reaction gas was switched to 3.2% H₂O/He. The water molecules absorbed in the zeolite further reacted with Cu-CH₃ sites to produce CH₃OH (see the Cycle 1 in Fig. R5). When the catalyst was

exposed in the reaction gas of $\text{CH}_4 + \text{H}_2\text{O}$ at $300\text{ }^\circ\text{C}$ for 15 min, Cu-OH group was formed on the 6MR. After the activation at $450\text{ }^\circ\text{C}$ in He (to remove the water molecules), the methane then was introduced to the Cu-OH sites at the reaction temperature of $300\text{ }^\circ\text{C}$. Methane reaction led to the formation of methanol precursors, which then was extracted via introducing 3.2% H_2O , the MS signals of methanol were shown in the following Fig. R5 (red color). It was found that methane reaction on Cu-OH sites produced more methanol than that on the isolated Cu sites, revealing a higher reactivity of Cu-OH sites. Therefore, Cu-OH site is more active in MTM.

Fig. R5. Methanol production on Cu-OH sites (Cycle 2, red color) and isolated Cu sites (Cycle 1, black color) in a stepwise MTM reaction at $300\text{ }^\circ\text{C}$. 100 mg $\text{Cu}_1/\text{SSZ-13}$ sample was firstly dried and activated in the flowing He at $450\text{ }^\circ\text{C}$ for 30 min, then methane was introduced into the catalyst when the temperature was cooled down to $300\text{ }^\circ\text{C}$, after 1 h-reaction, the catalyst was cleaned by the He flow, followed by the introduction of 3.2% water steam, and the signals of methanol were recorded by MS detector, this process was call as “Cycle 1”. The $450\text{ }^\circ\text{C}$ -activated $\text{Cu}_1/\text{SSZ-13}$ sample was exposed in the flowing reaction gas of $\text{CH}_4 + \text{H}_2\text{O}$ at $300\text{ }^\circ\text{C}$, after 15 min-reaction, the sample was purged by the flowing He and dried at $450\text{ }^\circ\text{C}$, then methane was introduced to the catalyst at $300\text{ }^\circ\text{C}$ and reacted with Cu-OH sites for 1 h, finally the same operations (He purge and methanol extraction with water) were conducted and the signals of methanol were collected, this process was called as “Cycle 2”.

3) The DFT looks nice, but there are rather high activation barriers, are there other possibilities that could be compared with? Own computations or others? What about coordination of more molecules like e.g. water? The calculated reaction path do not really prove anything without being discussed in a broader context.

Response: We thank the reviewer's comments. To support our predictable reaction cycle, we widely consider a sequence of possible DFT-derived pathways of the methane-to-methanol reaction as shown in Table R1. The pathway on Cu-OH located on 6-MR (energy profiles shown in Figure 5b) presents the lowest activation energy of 1.4 eV among the complete H₂O-oxidated reaction cycles that consider the transition state (TS) information in every step. Moreover, the evolution of the H atoms in CH₃OH and H₂ in this cycle is consistent with our experimental results in the isotope tracer experiment.

Table R1. Possible DFT-derived pathway of MTM reaction

Active Site	Zeolite	Oxidant	RDS ^a	Reaction Equation	TS(CH ₃ -H) ^b	TS(H-H) ^c	Ref.
CuOH@6MR	CHA	H ₂ O	H ₂ formation	CH ₄ +H ₂ O→CH ₃ OH+HH	0.6	1.4	This work
CuOH@8MR	CHA	H ₂ O	H ₂ formation	CH ₄ +H ₂ O→CH ₃ OH+HH	0.5	1.8	This work
Cu@6MR	CHA	H ₂ O	H ₂ formation	CH ₄ +H ₂ O→CH ₃ OH+HH	0.7	1.7	This work
CuOCu@8MR	CHA	H ₂ O	H ₂ formation	CH ₄ +H ₂ O→CH ₃ OH+HH	0.7	1.7	This work
CuOH@6MR	CHA	H ₂ O	C-H activation	CH ₄ +H ₂ O→CH ₃ OH+HH	2.1	1.4	This work
CuOCu-1	MOR	H ₂ O	H ₂ formation	CH ₄ +H ₂ O→CH ₃ OH+ H ₂	0.8	1.7	[1]
CuOCu-2	MOR	H ₂ O	H ₂ formation	CH ₄ +H ₂ O→CH ₃ OH+ H ₂	0.6	1.8	[1]
CuOCu	MAZ	H ₂ O	H ₂ formation	CH ₄ +H ₂ O→CH ₃ OH+ H ₂	0.8	1.7	[1]
Cu@6MR	MFI	H ₂ O ₂	OCH ₃ formation	CH ₄ +H ₂ O ₂ → CH ₃ OH+H ₂ O	1.2 [*]	2.0 ^{d,*}	[2]
CuOH@6MR	CHA	O ₂	C-H activation	CH ₄ +1/2O ₂ +H ₂ O→ CH ₃ OH+HHO	0.7	0.0 ^e	[3]
CuOCu@6MR	CHA	O ₂	O-O activation	CH ₄ +1/2O ₂ +H ₂ O→ CH ₃ OH+HHO	-0.3	-0.1 ^e	[3]

Cu@6MR	CHA	O ₂	C-H activation	Cu+CH ₄ →Cu-CH ₃ +H _b ^f	0.9	N.A. ^h	[3]
CuOCu	MFI	O ₂	C-H activation	CuOCu+CH ₄ → CH ₃ OH+Cu ₂	0.5(0.9) ^g	N.A. ^h	[4]
Cu ₂ (O ₂)	MFI	O ₂	C-H activation	Cu ₂ (O ₂)+CH ₄ → CH ₃ OH+CuOCu	1.7(1.5) ^g	N.A. ^h	[4]
CuOH	CHA	H ₂ O	C-H activation	CuOH+CH ₄ +H ₂ O→ CH ₃ OH+CuOHH+H _b ^f	1.1 [*]	N.A. ^h	[5]
CuOCu+Na	MOR	H ₂ O	C-H activation	CH ₄ +H ₂ O→CH ₃ OH+ H ₂	N.A. ^h	N.A. ^h	[6]
CuOCu	MOR	H ₂ O	C-H activation	CuOCu+CH ₄ +4H ₂ O→ CH ₃ OH+H ₂ +Cu ₂ O ₄ H ₆	0.8	N.A. ^h	[7]

^a: Rate-determining step; ^b: The relative energy of the transition state in C-H activation; ^c: The relative energy of the transition state in H₂ formation; ^d: The relative energy of the transition state in OCH₃ formation; ^e: The relative energy of the transition state in O-O activation; ^f: H_b refers to Brønsted acid site; ^g: Triple-state energy in the bracket; ^h: Not available; ^{*}: Gibbs free energy.

The reviewer also mentioned a possible pathway in which more molecules (e.g. H₂O) coordinate at the Cu site. The max coordination number of Cu is 4 in the intermediate M4 (see Fig. 5 in the main text). After one more water molecule is added into this system, this water can (i) dissociate in the CHA channel, (ii) connect with the Brønsted acid site via H-bond, or (iii) adsorb on the dissociated Cu site, which presents the adsorption energy of H₂O of -0.3, -1.0 or -0.7 eV, respectively (Fig. R6). However, H₂O in situations (i) and (ii) does not interact with Cu species and we do not think it can take effect on the reaction. The situation (iii) lacks evidence from the experiment and the literature. Thus, we have not investigated these situations in this work. The Cu site located on the CHA framework should not have the other void to adsorb more water molecules. Accordingly, Matěj Pavelka, et. al (J. Phys. Chem. A, 2006, 110, 4795) reported the maximum coordination number is 4 for Cu(I) complexes, which is consistent with our results.

Fig. R6. The possible DFT-derived structures and adsorption energies of the intermediate M4 in Fig. 5 after adding one more H₂O molecule.

Reference:

1. Mahyuddin, M. H., Tanaka, T., Shiota, Y., Staykov, A. & Yoshizawa, K. Methane Partial Oxidation over [Cu₂(μ-O)]²⁺ and [Cu₃(μ-O)₃]²⁺ Active Species in Large-Pore Zeolites. *ACS Catal.* **8**, 1500–1509 (2018).
2. Tang, X. *et al.* Direct oxidation of methane to oxygenates on supported single Cu atom catalyst. *Appl. Catal. B Environ.* **285**, 119827 (2021).
3. Sun, L. *et al.* Water-involved methane-selective catalytic oxidation by dioxygen over copper zeolites. *Chem* **7**, 1557–1568 (2021).
4. Yu, X., Zhong, L. & Li, S. Catalytic cycle of the partial oxidation of methane to methanol over Cu-ZSM-5 revealed using DFT calculations. *Phys. Chem. Chem. Phys.* **23**, 4963–4974 (2021).
5. Kulkarni, A. R., Zhao, Z.-J., Siahrostami, S., Nørskov, J. K. & Studt, F. Monocopper Active Site for Partial Methane Oxidation in Cu-Exchanged 8MR Zeolites. *ACS Catal.* **6**, 6531–6536 (2016).
6. Jeong, Y. R., Jung, H., Kang, J., Han, J. W. & Park, E. D. Continuous Synthesis of Methanol from Methane and Steam over Copper-Mordenite. *ACS Catal.* **11**, 1065–1070 (2021).
7. Sushkevich, V. L., Palagin, D. & van Bokhoven, J. A. The Effect of the Active-Site Structure on the Activity of Copper Mordenite in the Aerobic and Anaerobic Conversion of Methane into Methanol. *Angew. Chem. Int. Ed.* **57**, 8906–8910 (2018).

Figure 3,

a) There are two different lines labelled CH₄+H₂O. What does that mean?, d) That part is hard to read and assess.

Response: Thank you very much for pointing out this issue. In Fig. 3a, the two different lines labelled CH₄+H₂O indicate the spectra of the catalyst in the flowing CH₄+H₂O for different times (5 min and 10 min). We have added the reaction time in each label to clarify it, as shown below (Fig. 3). We also revised Fig. 3d and added some information in the revised manuscript.

Fig. 3. **a** *in situ* FTIR spectra of Cu₁/SSZ-13 catalyst in MTM at different temperatures (300-400 °C). **b** Steady-state FTIR spectra of Cu₁/SSZ-13 catalyst under different atmospheres at 400 °C. **c** Changes in the intensities of the selected IR bands in **a** and **b**. **d** *in situ* FTIR spectra of Cu₁/SSZ-13 during methane-steam conversion at 400 °C, the sample was firstly exposed in CH₄ + H₂O and then in He. Conditions: ~30 mg catalyst, total flow rate of 15 ml min⁻¹; 90% CH₄, 3.2% H₂O, He balance.

An important reference that is lacking is the one by Pappas et al. JACS 2017, 139, page 14961

Response: Thanks. The reference (JACS 2017, 139, 14961) reported by Pappas et al has been cited as the Ref. 16 in the revised manuscript.

Reviewers' Comments:

Reviewer #1:

Remarks to the Author:

The authors have revised the manuscript based on the reviewers' comments. One point made by the reviewer is that the oxygen concentration in the reactants should be quantified and reported. As indicated in the authors' response letter, they mentioned that some residual oxygen might be involved in the reaction ($2\text{CH}_4 + 2\text{H}_2\text{O} + \text{O}_2 \rightarrow 2\text{CH}_3\text{OH} + 2\text{H}_2\text{O}$).

Reviewer #3:

Remarks to the Author:

The new submission by Wang and coauthors after reviewers comments has raised the quality and soundness of the conclusions. Especially the new isotopic labeling experiments have given much stronger evidence for their claims and conclusions. In my opinion the manuscript can be published as it is.

Reply to the comments of the Reviewers

#Reviewer 1

The authors have revised the manuscript based on the reviewers' comments. One point made by the reviewer is that the oxygen concentration in the reactants should be quantified and reported. As indicated in the authors' response letter, they mentioned that some residual oxygen might be involved in the reaction ($2\text{CH}_4 + 2\text{H}_2\text{O} + \text{O}_2 \rightarrow 2\text{CH}_3\text{OH} + 2\text{H}_2\text{O}$).

Response: We thank the reviewer for the comment. According to the reviewer's suggest, we have quantified the oxygen concentration in the reactants by testing the oxygen concentration in water via MS. In the experiment, the deionized water in an airtight saturator was first heated up to 80 °C and kept at this temperature for 24 h in the flowing helium. Then, the appropriate amount of water (0.1-1.0 μl) was injected into MS detector in the flowing He. The signals of water and O_2 were recorded on MS and the results are shown in the figure below. It indicates that there was ~90 ppm O_2 in the water as the reactant. The related information has been reflected in the revised manuscript (see lines 448-449).

Fig. S22. The O_2 content in water. Prior to the measurement, the deionized water in an airtight saturator was heated up to 80 °C and kept at this temperature for 24 h in the flowing helium.